# From Leading to Guiding, Facilitating, and Inspiring: A Needed Shift for the 21st Century

C. June Maker 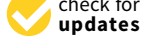

Department of Disability and Psychoeducational Studies, University of Arizona, Tucson, AZ 85705, USA; junemaker@hotmail.com

**Abstract:** In the 21st-century context, problem solving, creativity, critical thinking, collaboration, and communication are the most valued skills in the workplace. Thus, those in positions often labeled as "leadership" need to make a valuable shift: to *guiding, inspiring,* and *facilitating* rather than directing. In this article, I review research on two styles of leadership, *transformational* and *transactional*, and relate this research to discussions of the same two types of giftedness. Research on the effectiveness of leaders at engendering creative problem solving has shown the transformational style to be more effective. Leaders are guides in the *process* rather than the *content*, facilitators of the gathering and exchange of information from varied sources, and role models as they exhibit effective problem-solving behaviors themselves. As role models, they inspire others to take risks, think innovatively, and collaborate with others. Examples of methods for identifying exceptionally talented leaders and behaviors to observe are provided. In addition, an evidence-based model for igniting, cultivating, extending, and strengthening exceptional talent in leadership is described.

**Keywords:** leadership; gifted; creative problem solving; exceptional talent; collaboration; facilitating; inspiring; 21st-century skills

## 1. The Need for a Different Conception of Leadership and Talent in Leadership

In the 21st century, our increasingly interconnected world is facing unprecedented problems and situations that most people have never encountered or even believed could occur. Men and women in leadership positions have the potential to contribute to solving these significant problems, but they must do so in ways that are consistent with research on effective leadership and in recognition of the context in which those problems and challenges are occurring. In a recent IBM survey, 1541 chief executive officers in 60 countries and 63 industries identified creativity as the most valuable ability for top managers of the future [1]. These results correspond to the trend noted by economists, who observe a movement from managed economies to entrepreneurial economies [2], and who have identified four key "21st Century Skills": creativity, critical thinking, collaboration, and communication (http.//p21.org; accessed on 28 November 2021). These so-called "soft" skills are essential for leaders and groups who need to design new solutions for old problems and effective, innovative ways to meet challenges that have never been encountered or even imagined.

Leaders often are seen as people who have a vision and who are skilled at getting others to follow them to make this vision a reality. For example, in the Merriam-Webster Dictionary [3], synonyms for leadership include control, direction, management, governance, regulation, and surveillance. In other words, leaders think of the problems and solutions and others implement them as the leader directs, controls, and manages. What if this is an outdated and outmoded perspective? I propose a different view, one that fits with the current and future context: *effective leaders guide, facilitate, and inspire rather than direct or control*. They co-create with colleagues the vision that will guide their work together, and together define the problems, thinking of many methods for solving them and selecting the

best solutions to implement. The process becomes a *team* effort. Consistent with this view, leaders would be selected based on their 21st Century Skills of creativity, critical thinking, collaboration, and communication (http.//p21.org; accessed on 28 November 2021). One quality needs to be added: the *ability and willingness to inspire others to create positive changes in their lives and in their world*. Consistent with the vision they articulated together, leaders would collaborate with their teams to identify areas of need, help team members apply their critical thinking, give them the freedom to be creative, encourage free and open communication, and facilitate group decision making while inspiring everyone by manifesting these qualities themselves. In the academic literature on leadership, many different perspectives are offered and studied. Of all these perspectives, I believe the contrast between *transformational* and *transactional* leadership styles is the most important for the 21st-century context. The transformational style is the most appropriate for the current and future context, while the transactional style fits with the common perceptions of the past.

## 2. Transformational Leadership: An Important Style for the 21st Century

If all of us on this planet are to survive (and hopefully thrive), regardless of whether individuals are leaders of ideas or leaders of people, the focus of gifted leaders needs to be on changing the world to make it a better place, not just for themselves, but for other people, living beings of all types, and the plants that purify our air. Leaders also must give priority to solutions with long-term benefits, not just those that may solve the problem for now, but cause damage in the future [4–6]. In the literature on leadership, two styles or types have been identified: *transformational* and *transactional* [7–9]. Transformational leaders inspire and encourage those with whom they collaborate, thus motivating others to create positive change. Their focus is on designing a positive future for the organization or entity *in collaboration with the people whom they lead*. They lead by example rather than through a system of rewards and punishments or directives that need to be followed, leading to the personal growth of all involved, empowering and inducing self-belief in others. These positive role models focus on the *process* of creative problem solving rather than on a particular goal, solution, or method for implementing solutions and plans [10,11].

In an educational context, transformational administrators develop a shared vision [12,13] and identify, with teachers and other staff members, potential innovations that are consistent with this vision [14]. New programs are chosen based on the shared vision, agreed upon, and believed in by the majority of the staff [15]. The focus is on student outcomes rather than teacher or administrator innovations [12,16,17]. In a study involving four schools serving students in low-income areas, in two of the four schools, principals exhibited transformational leadership by involving teachers in making the decision to participate in the innovative program. In those schools, student achievement increased significantly [15,18]. In one school, for example, the average achievement scores on the Stanford Achievement Test (SAT-9) in reading, math, and language increased from approximately the 20th percentile to the 62nd percentile during the four-year period in which the DISCOVER team was involved [18]. Similarly, in a study involving exemplary principals in Australia [17], several principals described beliefs and practices from the perspective of transformational leaders. For example, they set up a guiding coalition to "help teachers build on their strengths" [17] p. 109, developed a shared vision and strategy by understanding that "The vision can only be a vision if the whole school is brought into it" [17] p. 109. These principals also led by building and sharing knowledge and information through modeling and being actively involved with teachers in the learning process. They acknowledged teachers' successes, recognized their accomplishments, and worked toward incorporating changes into the school culture [17]. Practices such as these empower teachers and help to embed new methods in the school culture, enabling staff to implement them more effectively while supporting each other [19,20].

Extending this idea to a more collaborative level means involving not only the teachers, administrators, and other staff members at the school, but also collaborating with

the students, their parents and caregivers, and individuals in related organizations in the school community [21]. One exemplary principal in New South Wales is an inspirational model for this practice, as she united the school community and the wider community of parents and influential members of the local area in the decision to bring in a new program, and, after field testing it, to implement it school-wide. As a result of school-wide implementation, and because the innovation was embedded into the school culture, teachers made significant changes in their practices [22] and expressed perceptions consistent with the underlying principles of the new teaching approach [23]. An important finding of these two studies was that teacher fidelity of implementation of the innovation was much higher than normally found in such studies [22]. Student growth in creative problem solving in math [24] and science [25], as well as in understanding the complexity of concepts and their interrelationships [26], was significant. Furthermore, the changes were more pronounced in the classrooms of teachers who implemented the model with a high level of fidelity, showing that the new teaching methods were key components of student growth.

On the other hand, *transactional* leaders use systems of rewards and punishments to exert "control" over the goals and solutions, as well as the methods used for implementing plans and solutions [7–9]. Incentives or benefits are awarded by transactional leaders for valued behaviors or results and penalties are imposed on those who do not follow guidelines or who do not achieve the desired results. Usually these systems are complex, but the goal is to ensure that the leader's agenda is followed, often rewarding the pursuit of given short-term goals [4,27]. Administrators in the two schools in Maker's studies [15,18], in which no overall increases in academic achievement were found, exhibited transactional leadership through actions such as selecting programs to implement without involving the teachers in decision making but monitoring their implementation of these projects. They used evaluation strategies designed by the program developers rather than designed cooperatively by teachers and administrators at the school. Administrators did not participate in the professional development provided to teachers, thus leaving evaluation and guidance up to outsiders who designed the programs and who were not familiar with the school contexts or the characteristics of the teachers. This practice also prevented administrators from modeling desired behaviors and approaches or finding appropriate ways to support teachers in implementing the innovative programs school leaders had chosen and were advocating. Neither the local community nor the students were involved in the decision-making process.

## 3. Transformational Giftedness: An Important Type of Giftedness for the 21st Century

An interesting parallel is in the recent writing of Sternberg [6,28]. He separates gifted people into the same two types, presenting evidence that methods for defining giftedness and identifying gifted people now and in the in the past have resulted in selecting those who are transactionally gifted. Transactionally gifted individuals are "identified as gifted and then expected to do something in return" [6] p. 231 such as getting good grades, performing well in special coursework, going to a prestigious school, and, most importantly, succeeding later in life by becoming eminent [29]. Sternberg argues that a new definition and new methods are necessary to fit the current world context. We need to identify and cultivate giftedness that is transformative, "that by its very nature seeks positively to change the world at some level—to make the world a better place" [6] p. 231. He further expands the concept to include both self- and other-transformation [28]. This is not a new concept. In Indigenous cultures, an underlying principle is the "connectedness of all life and a connection of all beings with the earth. As individuals and members of diverse cultures, we have a responsibility to protect 'Mother Earth' and to respect each other as spiritual beings with worth and significance regardless of our differences in characteristics, experiences, beliefs, and values" [30] p. 5.

In the Navajo culture in the USA, family members are expected to identify the gifts of each individual and nurture them [31]. In turn, "each individual is expected to use her or his gifts for the good of all" [30] p. 5. Similarly, in the Māori culture in New Zealand,



all students, especially those who are gifted, are expected to "give back" to their local communities. "Their responsibility as gifted individuals is to use their gifts to benefit all, not just themselves" [30] p. 5. This perspective is important to the island as a whole, and is embedded in curriculum standards in schools. In our mainstream culture, particularly at certain periods, this perspective has been articulated, but perhaps was not pervasive. Professor Virgil Ward often said that the difference between gifted people and those who are not gifted is that gifted people want to change their world to make it a better place, while other people want to simply understand and adapt to it. I suggest, at this crucial time, we need to return to the wisdom of using our gifts and talents for the good of our world.

Transformationally gifted individuals focus on making a positive and transformative difference, focusing on the *positive changes* that result from their actions *as their rewards*. On the other hand, although transactionally gifted individuals may make positive changes, their focus is on personal benefits (e.g., extrinsic motivation), such as money, recognition, or status that may result from their actions. Although a distinction often is made between intrinsic and extrinsic motivation, Amabile [32] makes a case for synergy between the two types of motivation. Synergy happens when "strong levels of personal interest and involvement are combined with the promise of rewards that confirm competence, support skill development, and enable future achievement" (p. 18). These are the kinds of rewards given by leaders with a transformational style. People are not born as one or the other type, but develop these tendencies through their interaction with mentors, role models, and other aspects of their environments [33]. In his argument for a paradigm shift, Sternberg poses an important question for educators of the gifted: "Do we want to continue to teach gifted children that being gifted means getting better grades or accelerating in one's studies, when there are so many problems in the world begging for solution?" [6] p. 232.

## 4. Exceptional Talent: A Conceptual Framework

The paradigm shift advocated by Sternberg [6] is similar to that advocated by Maker [5,34], in that its emphasis is on solving complex and varied problems for the good of the world rather than on gathering information, getting good grades, and performing beyond one's grade level. In Maker's definition [5,15,34,35], the term *exceptional talent* is used instead of gifted to situate it within the talent development paradigm [36]. Exceptional talent is defined as consisting of three interacting components: (a) the ability and willingness to solve the most complex problems, (b) the ability and willingness to solve a variety of types of problems, and (c) a highly integrated and interconnected knowledge structure within or across disciplines. Further included in this definition are adjectives to use when making judgments about the problem-solving processes and results. They need to be effective, efficient, economical, ethical, and/or elegant. Elegant is a criterion not always used; however, it is important for integrating the concepts of originality and novelty with appropriateness, as in definitions of creativity: "pleasingly ingenious and simple" [5] p. 162.

Not all adjectives may be appropriate for all solutions. For example, a solution may be efficient, but not effective, and another solution may be effective, but not efficient; another may be effective, but not economical, or economical, but not effective. In these cases, problem solvers must decide which are the most important criteria to apply in a particular instance. However, I argue that the ethical criterion needs to be applied in all cases, but in recognition of differing belief systems, while still following universal ethical principles, such as compassion, respecting the rights and needs of others, kindness, refraining from stealing or harming other people's property, and honesty [30,37].

Wisdom is necessary for problem solvers to know what attributes are needed for solutions to meet the four conditions of action that will change the world in beneficial ways, and is the overarching purpose of education for exceptionally talented individuals [5,6]. The third component of the definition, a rich, diversified, associative network of knowledge, facilitates creativity and problem solving [38–40]. Merging the definitions of transforma-

tional leadership and transformational exceptional talent leads to a conceptual framework for defining *exceptional talent in leadership*.

## 5. Exceptionally Talented Leaders: The Conceptual Framework Modified for Leadership

As in the framework proposed by Maker [5], described above, and as with other areas, *exceptional talent in leadership* has three interacting components: solving complex problems, solving varied types of problems, and having a complex and interconnected knowledge structure. However, exceptionally talented leaders are not only able to solve problems, but also to guide, facilitate, and inspire others to create solutions that are effective, efficient, economical, ethical, and/or elegant. Thus, exceptional talent in leadership has *four* interacting components. Leadership is a type of exceptional talent that often is overlooked by teachers [41], but is a talent essential for the success of groups and organizations. Ten human abilities have been outlined by Maker [5]: auditory, bodily/somatic, emotional/intrapersonal, linguistic, mathematical, mechanical/technical, moral/ethical/spiritual, scientific/naturalistic, social/interpersonal, and visual/spatial. Leadership talent differs from these. Unlike the other abilities, which are mostly connected to domains and symbol systems, exceptional talent in leadership cuts across and integrates several domains. For instance, three core abilities are necessary for transformational leadership regardless of the occupation, organization, or agency: social/interpersonal, emotional/intrapersonal, and moral/ethical/spiritual. Social/interpersonal abilities are essential for engendering cooperation, guiding groups and individuals, and inspiring them. Emotional/intrapersonal abilities enhance the ability of leaders to understand and manage their own emotions and reactions so they can be inspirational models for others. Moral/ethical abilities are essential for guiding others and themselves to make wise decisions that have impacts in local, regional, national, and international contexts in both the short and long terms.

Other abilities needed for exceptionally talented leaders may be different depending on the type of leadership position. For instance, in business and industry, leaders may need mechanical/technical, scientific/naturalistic, and mathematical abilities, while in education, linguistic abilities are important; in the arts, visual/spatial, auditory, and bodily/somatic may be important to enable the leader to understand and facilitate problem solving. In the culinary arts, bodily/somatic, with its inclusion of taste, would likely enhance a leader's ability to facilitate, guide, and inspire team members. In all these cases, however, being exceptionally talented in the three core areas is essential, while being exceptionally talented in the supporting areas is helpful, but not essential.

### 5.1. Exceptionally Talented Leaders and Creative Problem Solving

In the 21st-century shift from managed economies to entrepreneurial ones [2], increasing emphasis needs to be placed upon "making and using knowledge" [2] p. 303. In this new economy, employees, teams of managers, consumers, and others with an interest in a particular agency or organization will need to collaborate: share knowledge, create new knowledge, and use their knowledge in creative and innovative ways [42]. What is the role of exceptionally talented leaders in the creative problem-solving process? Consistent with their style of transformational leadership and the research on effective group problem solving, these guides, facilitators, and inspirers shape a climate of psychological safety and reflexivity (examining one's own feelings, reactions, and motives, leading to an understanding of the ways in which these factors influence what one does or thinks in a situation) [11]. The concept of reflexivity, as used in Carmeli and colleagues' research, corresponds to the definition of emotional/intrapersonal ability in the Prism of Learning [5], and is one of the three core components of leadership talent. Transformational leaders play a critical role in the workplace by providing support and motivating employees to engage in and display creativity [42–51]. Transformational leaders inspire and help others transcend their own self-interests and pursue collective goals, thus becoming effective beyond their own expectations [52]. Transformational leaders encourage their teams to challenge the

norm, question their assumptions and the old ways of doing things, and take risks by addressing problems and doing things in a novel way [53–55]. Two ways leaders do this is by being role models [11,42,56] and recognizing the value of the contributions of their team members [42,48,57,58].

In one of the key studies that have been conducted since 1982, Basadur [42] experimented with methods for leading others to think innovatively together, and found that a key component of success is for the leader to focus on the *process* of problem solving, rather than the *content*. Leaders transfer ownership of challenges by interacting with others as a process leader or coach; their job is to help everyone in the group "work together toward a useful solution" [42] p. 111. They model process skills and encourage others to use these same skills, thus becoming consultants or facilitators in the problem-solving process, rather than "giving orders" or solving the problems themselves. One practical method resulting from Basadur's research is a circular process for creative problem solving with four steps: generating, conceptualizing, optimizing, and implementing. Individuals have preferences and strengths at different steps; therefore, more effective teams can be assembled if individuals with different strengths and preferences are expected to work together. They may experience more difficulties in the beginning, but after they have accepted and begun to respect their differences, the results are superior to those of teams whose members have the same strengths and preferences. The instruments designed by Basadur [42] and Treffinger and colleagues [59] can be used to assess individual styles and preferences. Another method is the creative profiler [39], which is based on creativity assessments, and includes strengths in cognitive (e.g., divergent thinking, mental flexibility) and conative (e.g., risk-taking, openness) factors. The creative profiler can be used both to assist individuals in developing strengths and meeting challenges and also to match them with others with similar and different profiles—for the benefit of the individuals and their collaborative groups.

### 5.2. Collaborative and Individual Creative Problem Solving

In the past, a prevailing belief was that individuals produced more creative solutions than groups, especially during brainstorming (c.f. [60,61]). However, the writing and research of Amabile and colleagues (c.f. [32,62]) has revealed a different way of viewing the work of individuals and groups when solving problems creatively. Consistent with Amabile's ideas and research, Brophy [10] separated problem-solving tasks into single-part problems and multi-step problems in his review of research and in his own investigation. In a review of 92 studies of individual–group comparisons, individuals did better in 83% of the 60 *single-part* tasks, groups did better on 7% of these tasks, and they did equally well on 10% of these tasks. However, with the 15 *multi-part* tasks, one individual (7%) did better, individuals and groups performed at the same level on three (20%) of the tasks, and on 11 (73%) of the tasks, groups performed better [10]. In his own study, Brophy investigated the creative outcomes (number of solutions, originality of solutions, quality of solutions, cost–benefit trade-offs and reductions, meeting all task demands, addressing all problem parts, selecting solutions, and defending their choices) of the same 51 interactive (group) and nominal (single-person) problem-solving groups. He found that when working on a multi-part task, the same 51 interactive groups did better than the same 51 nominal groups, and the effect sizes were large ($F$ [1, 50] = 731.59, $p < 0.001$, $d = 4.78$).

In the current context, another useful definitional shift has been to view creativity as *both a process and a product*. Although some believe that the creative process is more important than the product (c.f. [63]), most definitions have been focused on the attributes of the product (c.f. [39,57,64–66]). An integrated perspective is to acknowledge the importance of creative processes for producing creative outcomes [59,62]. For example, regardless of the methods used to score originality, studies have shown that the more ideas one produces (fluency), which results from the creative processes of divergent thinking and deferred judgment [67], the more likely one is to produce original ideas, the quality deemed to be most important in creative products [68]. In studies of creative problem solving, the creative

process can be defined as "a process whereby individuals are able to identify and construct a problem, engage in information search and encoding, discover, evaluate, and select the most novel solution" [11] p. 117. This integrated perspective is particularly relevant when considering the creative problem solving of groups and the leadership needed to guide, facilitate, and inspire.

*5.3. Problem Types and the Discover Framework*

Research and practice emanating from Discovering Intellectual Strengths and Capabilities while Observing Varied Ethnic Responses (DISCOVER) has built upon and extended the original framework of problem solving as a key component in giftedness [34]. In this framework, the seminal work of Getzels and Csikszentmihalyi [69,70] was modified to include three fundamental types of problems [5], based on the components of problems and the amount of information available to both the problem solver and the one proposing the challenge: the problem, the method, and the solution. The first type, *closed*, has a well-defined problem, a specified method, and one acceptable solution; the problem solver has to apply the given method and identify the solution known by the presenter of the problem. The second type, *semi-open*, has a well-defined problem, a *range* of acceptable methods, and a *range* of appropriate solutions. The third type, *open-ended*, has a problem situation that is either partially defined or completely undefined, an unlimited number of possible methods, and an unlimited number of appropriate solutions [5,35]. This is the most complex type of problem, and the kind defined by Brophy [10] as a multi-step problem. This kind of real-world, complex problem is the type studied by Carmeli and colleagues [11,56] in their investigation of the impact of transformational leadership on creative problem solving. Similarly, Basadur [42], in studies of creative leadership, has used the term *applied creativity*, which is a process necessary for finding and solving complex problems that involves having an actual creative product or plan as the final result [42] p. 104.

## 6. Identifying Potential and Actualized Exceptionally Talented Leaders

If we have such a great need for these exceptionally talented leaders, how do we identify those with the greatest potential or those who have actualized their talents so their development can be nurtured or they can be chosen for leadership positions? Over the years, in the DISCOVER projects [15,71–77], we have experimented with assessments of creative problem solving related to the three core abilities of social/interpersonal, emotional/intrapersonal, and moral/ethical/spiritual [5]. For instance, in the first versions of the assessments for grades K–2, 3–5, and 6–8, we noted behaviors indicating interpersonal (social) abilities and intrapersonal (understanding of self) behaviors while students were interacting in groups to solve individual problems. Later, we designed an assessment for high school students (grades 9 to 12) in which three to five students solved a problem together, making a large triangle with tangrams. Recently, we designed separate assessments of all ten abilities in the Prism of Learning model for very young children (ages 4 to 6) and field tested them in both the USA and the United Arab Emirates [78]. The assessment battery included the three core abilities (social/interpersonal, emotional/intrapersonal, and moral/ethical/spiritual), which were combined into a leadership cluster.

As part of the development and field testing of all the performance-based assessments, the research team asked observers (one for each group of three to five students) to identify students whom they considered to be superior problem solvers in a particular domain, and then to tell the researchers what the students did or said that led to this belief (observable behaviors). Observers were from various age groups, different cultures, different occupations, and different perspectives [79]. After approximately 5000 students were observed, checklists of these behaviors were provided for future observers. Observers also could add behaviors not on the checklist, and the research team periodically examined these "write-ins" to determine if other behaviors needed to be included [15,74]. In Table 1, I have listed the behaviors identified during all levels of assessments that could be considered

to be indicators of transformational or transactional leadership potential. Some could be indicators of either or both, depending on the situation.

**Table 1.** Indicators of Potential or Actualized Transformational and/or Transactional Leadership Styles.

| Domain or Type of Task | Transformational | Transactional | Either or Both |
|---|---|---|---|
| **Social/ Interpersonal** | Evaluates others by making positive comments or giving constructive criticism<br>Shows pleasure when others solve a problem or complete a task<br>Encourages others to attempt difficult tasks<br>Assumes responsibility for writing, recording, or facilitating the activity<br>Assists observer in accomplishing his/her purposes<br>Attempts to involve all group members in activity<br>Keeps mood of group cheerful<br>Makes suggestions to group without dominating<br>Changes behavior if it seems to affect the group in a negative way<br>Manages own and/or group disappointment effectively<br>Finds ways to include others<br>Takes turns | Others follow his/her suggestions<br>Competes with others<br>Organizes group activity | Shows humor when interacting with others<br>Initiates sharing of product with others<br>Stories/pictures/constructions demonstrate understanding of emotions and motivations of others<br>Stories/pictures/constructions demonstrate understanding of social relationships<br>Demonstrates interpersonal behaviors valued by own culture<br>Demonstrates interpersonal behaviors valued by dominant culture<br>Linguistic product includes self as a character |
| **Emotional/ Intrapersonal** | Competes with self<br>Demonstrates confidence in self<br>Comments about own abilities are consistent with observed abilities<br>Describes actions demonstrating effective management of emotions | | Enjoys solving problems or completing constructions<br>Stories/pictures/constructions demonstrate understanding of emotions and motivations of self<br>Identifies own emotions<br>Identifies reasons for emotions<br>Describes behavior consistent with emotions identified<br>Identifies different emotions for different situations |
| **Moral/ Ethical/ Spiritual** | Talks about fairness<br>Talks about love<br>Talks about honesty<br>Talks about sharing<br>Talks about caring for the natural world | Talks about hitting or hurting others | Talks about correcting wrong behavior |

**Note:** These behaviors were observed when students were working together, either to solve individual problems in a group setting or to solve a problem as a group during the Discovering Intellectual Strengths and Capabilities while Observing Varied Ethnic Responses (DISCOVER) performance assessments.

Interestingly, young people of all ages exhibit behaviors that can be indicators of one or the other style of leadership. In the DISCOVER assessment of social/interpersonal abilities for children ages 4, 5, and 6, for example, children were given large blocks of different sizes and shapes along with a picture of a bridge that can be made with the blocks. Their task was to *build one bridge, all together*. During the first trial of the assessment, I saw two young children who were already exhibiting transactional and transformational leadership styles! Most of the children in the group had built a very nice bridge together. Then, a little girl put a block in the middle of the bridge and refused to move it even when the other children told her it was blocking the traffic going over the bridge. She insisted it was important in the design of the bridge. When the children seemed to be at an impasse, one little boy turned the block sideways and said, "it can be a car going across the bridge." Everyone was pleased and satisfied, including the girl who had placed it there. He was definitely a transformational type of leader in the making! The little girl had her own agenda and was intent on having her own way, showing a tendency toward a transactional style.

Emotional/intrapersonal abilities have been observed in separate tasks, as in the DISCOVER assessment of young children, or as an aspect of other activities. For instance, after high school students made the large triangle, they were asked to reflect on their participation in the group, including how they had helped and/or hindered the work of the group. Being able to identify their own strengths and weaknesses, as well as being able to manage their emotions, was an important aspect of these observations. Young children

identified emotions and then explained what they would do (and why they would do it) if they experienced a particular emotion.

Moral/ethical/spiritual abilities have been included only in the assessment of young children: they said what they would do if they saw negative events happening, and then described and drew a good person. In these assessments, children expressed positive ethical values that seemed far beyond what one would expect for their ages. One 5-year-old drew a picture of the world with people of different colors of skin and hair, and said "All people are good people. They have different hair and skin, and they live in different places." Children, regardless of their cultures, talked about positive ethical values, such as kindness, love, and honesty (Table 1). They were showing their potential to develop the ethical values needed for wise use of their talents as leaders.

Reflecting on these results and the lists of behaviors in Table 1, one can see that behaviors indicative of a transformational style have been identified more frequently by observers. Why? One possible reason is that children have a tendency toward transformational styles, developing transactional ones after experiencing systems of rewards and punishments in school and at home. Because observers were asked to focus on positive behaviors rather than negative ones, a second possibility is if observers had a transformational view, then perhaps they identified behaviors from this perspective. A third possibility is the types of tasks presented to students stimulated them to act in ways more characteristic of a transformational style. Another reason could be a cultural component. As noted earlier, Indigenous groups, such as Navajo, Tohono O'Odham, and Yaqui, as well as Hispanic cultures, which were prevalent in the schools involved in the DISCOVER assessments, tend to be more collaborative in their activities, and thus demonstrate a more transformational style of leadership. In the DISCOVER high school assessment, for instance, of the 303 students in one study, 50% were Hispanic, 29% were Navajo, and 20% were White [80]. In another study, 47% were Navajo, 38.5% were Hispanic, and 14.5% were White [76]. In a third study, 44% were Navajo, 46% were Hispanic, and 20% were White [75].

## 7. Igniting, Cultivating, Extending, and Strengthening Exceptional Talent through Real Engagement in Active Problem Solving

Regardless of whether young people have demonstrated potential for exceptional talent in leadership, but especially if they have, Real Engagement in Active Problem Solving (REAPS) is a valuable teaching approach that can help to develop exceptional talent in leadership in both general and special classrooms at all levels of education [5,21–23,81]. Since developing the conceptual framework for defining exceptional talent [34], the DISCOVER researchers have experimented with different methods for developing problem-solving abilities. In 2008 [82], we designed an approach that we have found to be successful based on field testing and evaluation at different grade levels and in various cultures [5,22,23,83–85]. The REAPS model is an integration of four evidence-based teaching models with problem solving as a common goal (Figure 1): the DISCOVER problem types and curriculum principles [15], the Thinking Actively in a Social Context (TASC) problem-solving process (c.f. [86,87]), the Problem Based Learning (PBL) approach to selecting real-world problems and solving them from the perspective of different stakeholders (c.f. [82,88]), and the Prism of Learning domains of ability [5]. All of the models require and develop the 21st Century Skills of creativity, critical thinking, collaboration, and communication.

**Figure 1.** Real Engagement in Active Problem Solving (REAPS).

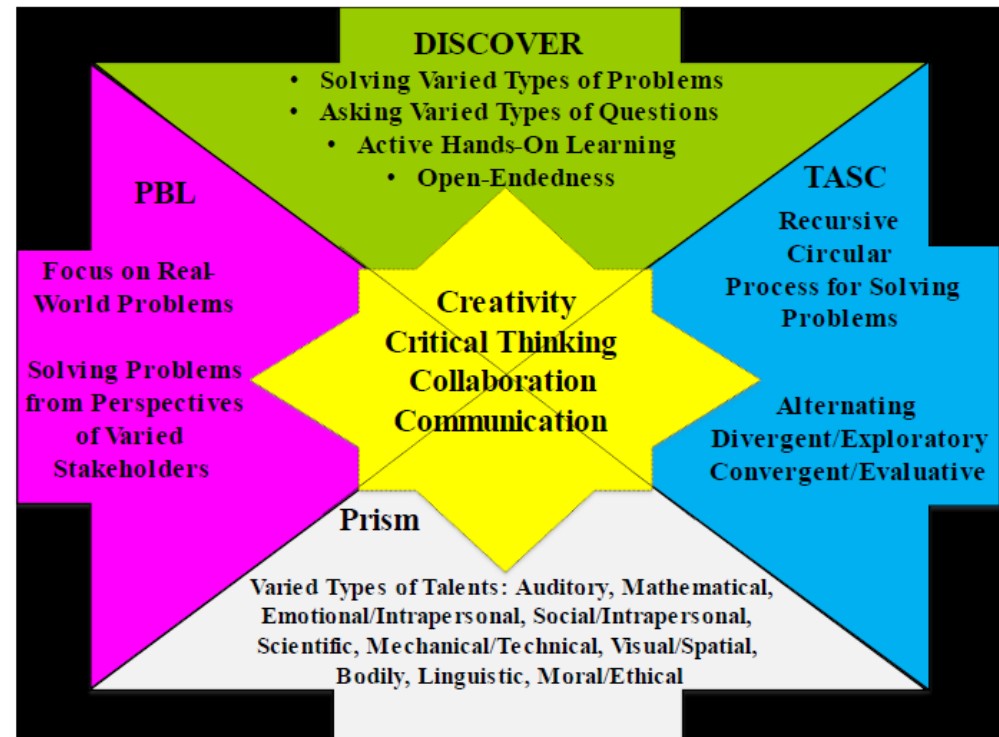

**Note**: The four models that make up REAPS all contribute to the development of key 21st Century Skills. The models also provide methods for participants and guides to use during the problem-solving process.

When teachers use the REAPS model, they choose local, national, and international real-life problem situations that are developmentally appropriate for the students. After students have experienced the model, they are invited to choose problems they believe are significant. For instance, in Grade 1 classrooms in Australia, children decided to solve the problem of loneliness on the playground. Their solutions showed clearly their focus on making their world a better place for children at the school: a loneliness bench where children who were lonely could come and someone would find others to play with them and a "loneliness cop" who went around the playground looking for children who seemed lonely, asked them if they wanted to play with someone, and found a child or group of children who welcomed them [22]. At other levels, students either chose or were introduced to problems, such as whether to build a resort in an indigenous rainforest [22], how to reduce or reverse the effects of desertification in the arid land in which they live [30], and how to reduce the ocean pollution that is harming the marine life the local residents depended upon for food [83,84].

To assess and develop the third component of giftedness, the rich, associative network of knowledge essential for creativity and problem solving [38,39], and to understand the complexity and interrelationships among concepts [5], we have used concept maps for pre-and post-assessments [26,89–91]. In these cases, the concept maps were used to assess students' structure of knowledge related to the content of their problem solving. However, if one is assessing their potential for exceptional talent in leadership, the concepts students would be asked to map would be related to the core abilities that make up the leadership cluster. For example, they could be given concepts such as evaluate, inspire, compete, make positive comments, collaborate, guide, manage, control, respect, role model, creative problem solving, encourage others to be creative, influence, power, and other similar words. Concepts could be taken from descriptions of both types of leaders. A focus question to propose for the mapping exercise could be "Who is an effective leader?" More information

about concept mapping and its use in identifying exceptional talent can be found in articles by Maker and Zimmerman [5,35,89].

Most recently, the DISCOVER team collaborated with the United Arab Emirates University [92], teaching an online class on creative problem solving for high-potential high school students in the UAE. We put forth a significant international problem that is a national priority in their country: plastic pollution. We first administered concept maps related to life science and climate change to help us make a pre- and post-assessment of changes in their understanding of the complexity and interrelationships of concepts related to the problem they were solving. With teachers as facilitators of small groups and the DISCOVER team as guides in the process, students followed the TASC steps of gathering and organizing information, identifying the problem, generating ideas for solving it, deciding on the best solution(s), developing a plan for implementing the solution(s), evaluating the solution(s), communicating the results to a real audience, and reflecting on their learning. This process is an effective way to reach creative solutions because it alternates between divergent-exploratory and convergent-integrative thinking [93] and is a method everyone can learn and continue to apply in many areas of their lives [87]. The students in the class were in stakeholder groups that were real in their country or created for the purpose of having variety in the interests of the groups: waste management, the plastic industry, a student-led group working for a plastic-free Abu Dhabi, and the government agency responsible for environmental protections.

Students' solutions showed their passion for creating a better world for both people and the marine animals in the ocean near their community. In their group solutions, most students demonstrated a transformational type of exceptional talent. Several groups and individuals also exhibited transformational types of leadership. To create their solutions, they surveyed community members, presented potential solutions to audiences, and incorporated responses from community members into their final solutions. One example was the plastic industry group. They asked community members what substances could be substituted for the materials normally used that would be more eco-friendly. The majority of respondents suggested bamboo, so the group researched what products could be made from bamboo and incorporated this as their most important solution.

These experiences and our research on the results of implementing the REAPS model show it can be an effective way to develop transformational forms of exceptional talent, and also can be effective as a context for igniting, cultivating, extending, and strengthening student potential for developing exceptional talent in leadership. When the teacher acts as a process coach, guiding and facilitating the work of the group rather than directing the group's decision making [22,23], the teacher is acting as a role model, prompting students to employ this important leadership style in their problem solving in school and as professionals.

## 8. Conclusions

Solving the unprecedented, complex problems threatening our way of life and the ecosystems in which we live will require transformational leaders who themselves possess the 21st-century skills of creativity, critical thinking, collaboration, and communication. We need these exceptionally talented leaders, not to solve the problems themselves, but to guide, facilitate, and inspire others to develop and use these skills by being an example and by guiding others in the *process* of solving problems. They are transformational rather than transactional in their style and their motivation. These inspirational leaders are motivated, not by their desire for personal rewards, such as power, influence, or money, but by their sincere desire to make the world a better place.

Young children and young adults already exhibit signs of exceptional talent in leadership through behaviors such as encouraging others, helping to resolve conflicts in group settings, understanding and managing their own emotions, and talking about positive ethical values, such as fairness, honesty, and sharing. If these characteristics are ignited, nurtured, and extended through Real Engagement in Active Problem Solving, in which

they collaborate with their peers and design solutions to real-world problems that are of concern to them, and are encouraged to design innovative plans they can implement, then they can actualize this potential and become the leaders we so desperately need.

The evidence-based models in REAPS contribute to different experiences that are related to the development of exceptional talent in leadership. The Prism of Learning helps students and teachers identify and develop diverse talents that are components of leadership abilities. The DISCOVER curriculum model helps teachers choose open-ended problems as the main focus for real-world problem solving and to present closed and semi-open problems within the overall focus on open-ended problems. Problem Based Learning contributes an emphasis on the real-world nature of problems with its corresponding inclusion of different stakeholders, which helps students understand how diverse interests and perspectives need to be considered and integrated into their solutions if they are to be successful as transformational leaders. When teachers guide their process and give them ownership of their solutions and the ways they implement their plans, students have a role model to follow that will enable them to become process coaches for their own collaborative groups in the future.

The most important conclusion I have reached from the 30 years of experimenting with performance-based assessments, the Real Engagement in Active Problem Solving (REAPS) teaching model, and other methods for identifying and nurturing exceptional talent is this: when considering exceptional talent in leadership, decisions must be made on the basis of *observations* in real-world situations, not on the basis of *self-reports* of leadership preferences and actions. Although what people *say* can be an indicator of their tendency toward a particular style, what they *do* is more important.

**Funding:** No funding was received.

**Institutional Review Board Statement:** This article does not contain data from research on human subjects unless the data were reported in another article, and if so, that article was cited.

**Informed Consent Statement:** Not applicable.

**Data Availability Statement:** Not applicable.

**Conflicts of Interest:** I have no conflict of interest.

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
