# Peer review of "From Leading to Guiding, Facilitating, and Inspiring: A Needed Shift for the 21st Century"

_education, doi:10.3390/educsci12010018_

Round 1

Reviewer 1 Report

Dear Author(s), 

Thank you for the opportunity to review your work. 

The work is interesting, and has potential. I see its potential value, and I also see a need for greater clarification in areas. 

The introduction proposes a number of 'future states', which are not unique to the literature. The articulation that "leaders have a vision and are skilled at getting others to follow them to make this vision a reality. In other words, they think of the problems and solutions and others implement them as the leader directs, controls, and manages. What if this is an outdated and outmoded perspective?" is inaccurate of the literature and of practice. Dominant models of leadership such as servant, authentic, ethical, virtuous, and empowering each offer a direct alternative to this model. I suggest that a stronger theoretical basis is needed for this, along with future assumptions being made. 

The work needs to be better situated in developing theoretical and practical outcomes that creates a clear theoretical model linking these concepts. At present I feel this work is largely a presentation of fact rather than a nuanced and clear analysis of the work. 

I wish you the best with your manuscript.

Author Response

Please see the attached response to reviewers. I attempted to submit the revised manuscript, but could not find a place to upload it. 

Reviewer 2 Report

The presented manuscript contributes to the understanding of education and leadership. However, the argumentation leading up to the conclusion is presented in a less-than-optimal manner.

The overall strength of the article is the in-depth use of literature to argue the conclusion. The conclusion alone is absolutely clear, easy to follow and justified by the paper. 

However, the major drawback is that some paragraphs are hard to follow, and, e.g. a clarifying method section does not exist. I had to read and re-read the manuscript several times, and  Examples of this: 

All headings are on the same level, which makes me wonder what is connected. 

Line 203: The section caption mentions a framework, but it is unclear what the framework is and how it contributes to talented leaders and problem-solving. 

Line 415: The description between the REAPS model, DISCOVER, TACS, PBL and Prism of Learning could be made more accessible in a figure or table.

The article does not seem to follow the MDPI reference style guide, which prescribed numbers in brackets. https://mdpi-res.com/data/mdpi_references_guide_v5.pdf

Author Response

Attached is my response to both reviewers. I attempted to upload a revised version of the manuscript, but could not find a place to put it.

Round 2

Reviewer 2 Report

The author has made several improvements, but some work still remains: The contribution by the four models are still rather blury, and if the author indeed mean that the paper doesn't need to clarify the connection between the models, this part of the paper could benefit from improved clarity and simplicity. 

Some of the changes have aided the overall clarity, so first part of the paper and the conclusion stands clear now. 

Author Response

Attached is a file with my response to the reviewer comments.

Round 3

Reviewer 2 Report

The paper has been improved for clarity, and the link between the relevant theories is underlined. Hence, I have no further comments.

Author Response

Thank you for your positive comments.

The only disagreement I have is with the comment about the need for minor English language and spellcheck. I have been teaching professional writing to doctoral students for 15 years, so I have been respected for my grammatical clarity and knowledge. I have reviewed the manuscript again and find no problems with the language and spelling. If you see some, please point them out to me, and I will correct them.
